# NEURAL NON-ADDITIVE UTILITY AGGREGATION

## ABSTRACT

Neural architectures for set regression problems aim at learning representations such that good predictions can be made based on the learned representations. This strategy, however, ignores the fact that meaningful intermediate results might be helpful to perform well. We study two new architectures that explicitly model latent intermediate utilities and use non-additive utility aggregation to estimate the set utility based on the latent utilities. We evaluate the new architectures with visual and textual datasets, which have non-additive set utilities due to redundancy and synergy effects. We find that the new architectures perform substantially better in this setup.

## 1 INTRODUCTION

In this paper, we study the problem of learning to predict utilities for sets of objects, which we denote *set regression*. Let $\mathcal{O} = \{o_1, \ldots, o_{\_}\}$ be a set of *objects*. Let $\mathbb{C} = \{C_1, \ldots, C_{\_}\}$ be a set of sets that consist of objects from set $\mathcal{O}$, i.e., $C_k = \{o_i, \ldots, o_j\}$ with $o_i, \ldots, o_j \in \mathcal{O}$. During training, models only observe sets from $\mathbb{C}$ and their corresponding utilities and have to learn a function $v : \mathcal{C} \to \mathbb{R}$ that maps from sets to set utilities. At test time, utilities of unseen sets with potentially unseen objects have to be estimated.

The standard idea to approach this problem in representation learning is to learn representations for sets in $\mathbb{C}$ such that the utilities can be predicted based on the generated representations. More formally, for sets $C_k \in \mathbb{C}$ an aggregation function $\bigcirc_{o \in C_k} \varphi(o)$ is learned that produces an aggregation of the individual object representations $\varphi(o)$. A function $v(\bigcirc_{o \in C} \varphi(o))$ is trained (usually jointly with the aggregation function) to predict the utility of the set. One often used aggregation strategy is to simply add the individual object representations $\varphi(o)$ Zaheer et al. (2017). Another strategy is to use recurrent neural networks (RNNs) that learn to aggregate object representations. On a conceptual level, the idea of both approaches is to encode everything that is relevant to predict the utility of a set in its aggregated representation. This approach, however, ignores the fact that meaningful intermediate utilities for the objects contained in the sets can be predicted in many problems. For example, intermediate utilities of sentences in automatic summarization can be predicted and aggregated to obtain the summary utility.

A crucial limitation of this idea is the fact that objects in sets do not necessarily have an intrinsic value that is independent from the other objects in the same set. For example, repeating the same information multiple times in automatic summarization does not improve the summary utility even though the individual sentences might contain crucial information. Furthermore, a sentence $A$ might depend on another sentence $B$ (e.g., to resolve pronouns) such that sentence $A$ can only be understood if sentence $B$ is also present in the summary. Even if sentence $A$ contains valuable information, it does not have an intrinsic value. Its value becomes only effective when $B$ is also included in the summary. Since such redundancy and synergy effects occur in many problems, additive utility aggregation is not appropriate.

To resolve this problem, we propose *neural non-additive utility aggregation*. Contrary to previous approaches, we propose to learn intermediate latent utilities $u$ for objects in $\mathcal{O}$ and a non-additive aggregation function for the intermediate utilities. To this end, we make use of the discrete Choquet integral, which is a non-additive generalization of the well-known Lebesgue integral. Instead of using additive measures, the Choquet integral makes use of non-additive capacities. Hence, it is well-suited to deal with phenomena such as redundancy and synergies.

$$\varphi_1 \oplus \varphi_2 \oplus \cdots \oplus \varphi_n \longrightarrow v$$

(a) DeepSets (DS)

$$\varphi_1 \qquad \varphi_2 \qquad\qquad \varphi_n$$
$$\downarrow \qquad \downarrow \qquad\qquad \downarrow$$
$$h_0 \longrightarrow h_1 \longrightarrow h_2 \longrightarrow \cdots \longrightarrow h_n \longrightarrow v$$

(b) Recurrent Neural Network (RNN)

$$\varphi_1 \qquad \varphi_2 \qquad\qquad \varphi_n$$
$$\downarrow \qquad \downarrow \qquad\qquad \downarrow$$
$$h_0 \longrightarrow h_1 \longrightarrow h_2 \longrightarrow \cdots \longrightarrow h_n$$
$$\downarrow \qquad \downarrow \qquad\qquad \downarrow$$
$$u_1 + u_2 + \cdots + u_n = v$$

(c) Recurrent Choquet Network (RCN)

$$\varphi_1 \qquad \varphi_2 \qquad\qquad \varphi_n$$
$$\downarrow \qquad \downarrow \qquad\qquad \downarrow$$
$$h_0 \longrightarrow h_1 \longrightarrow h_2 \longrightarrow \cdots \longrightarrow h_n$$
$$\downarrow \qquad \downarrow \qquad\qquad \downarrow$$
$$g_1 \cdot u_1 + g_2 \cdot u_2 + \cdots + g_n \cdot u_n = v$$

(d) Deep Choquet Regression (DCR)

Figure 1: Illustration of prior architectures (1a and 1b) and the newly proposed architectures (1c and 1d). $\varphi_i$ denote feature representations of the input data and $h_i$ indicate hidden states. $g_i$ are gates that are responsible for handling redundancy and synergy effects in the DCR architecture. Each arrow models a (potentially non-linear) function. The + sign indicates addition of latent utilities and the $\oplus$ sign indicates addition of vectors. $v$ and $u_i$ indicate the predicted set utility and predicted latent utilities, respectively.

**Main Contributions.** In this paper, (i) we propose two novel network architectures for set utility prediction that explicitly exploit the fact that many problems can be modeled as non-linear aggregation of latent element scores, (ii) demonstrate the superiority of the newly proposed architectures compared to traditional representation learning in computer vision and natural language understanding datasets, and (iii) make all datasets and code publicly available for further research.

## 2 NEURAL NON-ADDITIVE UTILITY AGGREGATION

In this section, we present two neural non-additive utility aggregation architectures based on the Choquet integral. The key idea implemented in both architectures is to train models that predict a latent intermediate utility for observed objects in a given set in each step. The latent utilities are aggregated with a learned non-additive aggregation function. This idea is inspired by the discrete Choquet integral Choquet (1954); Grabisch et al. (2009). The non-additivity of the Choquet integral is crucial for our work, because we focus on non-additive problems such as automatic summarization. Both architectures are described below. We illustrate both architectures along with prior approaches in Figure 1. Due to space restrictions, we refer to Tehrani et al. (2012a) for a solid discussion of the Choquet integral for machine learning and focus in this work on the newly proposed architectures and their evaluation.

### 2.1 RECURRENT CHOQUET NETWORKS

The first architecture, Recurrent Choquet Neural Networks (RCNs), uses an RNN-like approach to construct a hidden state in each step. The hidden state is used to predict a latent intermediate utility that already takes non-additive effects into account. The predicted utilities are directly related to the aggregation of capacities in the Choquet integral, which also take non-additive aggregation effects into account. Consequently, the predicted utilities can simply be summed to obtain a non-additive set utility. The architecture is illustrated in Figure 1c.

In this work, we model hidden states $h_i$ according to $h_i = tanh(W^{(1)}_{|\varphi| \cdot |h|, |h|} \cdot (h_{t-1} \parallel \varphi_t))$, where $\parallel$ denotes feature vector concatenation, $|h|$ denotes the size of the hidden states and $|\varphi|$ the size of the input representation. The latent utility at step $t$ is computed according to a simple linear layer $u_t = W^{(2)}_{|h|,1} \cdot h_t$. The set utility is then computed by $v = \sum_{i=1}^{n} u_i$. Needless to say, other, more sophisticated functions can also be implemented. In this work, we decided to focus on rather simple functions to focus on the essential novel part of the new architectures and to achieve a better comparability to the reference models.

## 2.2 DEEP CHOQUET REGRESSION

The second architecture, Deep Choquet Regression (DCR), predicts a utility for each object in the set individually. To model non-additive effects, hidden states are learned, which are used to predict gates $g_t$. Note that the gating in DCRs is different to the attention mechanism, since the attention mechanism aggregates representations and not latent intermediate utilities.

Again, the products of gates and predicted intermediate utilities can be viewed as capacities of the Choquet integral. The architecture is illustrated in Figure 1d. The computation of the hidden states equals to the RCN. The individual utilities are computed directly based on the object representations and not based on the hidden states. Hence, we obtain $u_t = W_{|\varphi|,1}^{(1)} \cdot \varphi_t$. The gates are computed according to $g_t = \sigma(W_{|h|,1}^{(2)} \cdot h_t)$. The set utility is computed according to $v = \sum_{i=1}^{n} g_i \cdot u_i$.

# 3 EXPERIMENTAL SETUP

In the following experiments, we want to evaluate whether the proposed approach based on the Choquet integral performs better than the standard representation learning approach in situations in which an aggregated score of multiple elements has to be estimated. Specifically, we are interested in situations in which the aggregated score cannot simply be estimated by adding individual scores but tasks in which the aggregation depends on previously observed elements. Next, we describe the datasets and the experimental setup that we designed to gain insights on this setup.

## 3.1 REFERENCE ARCHITECTURES

The goal of the experiments is to investigate whether aggregating intermediate results works better than predicting the utility based on a single representation. Hence, we compare the newly proposed architectures to two standard approaches in representation learning that both generate representations for sets.

### 3.1.1 SUM BASELINE

The Sum Baseline (SB) is inspired by prior works in group rating, in which set utilities are usually modeled as the sum of individual instance utilities. This baseline simply learns to predict instance utilities independently and sums all predicted utilities. Similar to the new architectures, we use a simple linear model in a first version of the model, namely SB-s (s for small), to predict element utilities: $u_t = W_{|\varphi|,1} \cdot \varphi_t$. In a second version, SB-l, we use a slightly more complex function: $u_t = W_{8,1}^{(1)} \cdot \sigma(W_{|\varphi|,8}^{(2)} \cdot \varphi_t)$.

This baseline is also a way to determine the strength of the non-additive effects. If the set utility is based on less non-additive effects, e.g. only small redundancy effects, the Sum Baseline should be able to yield a good performance.

### 3.1.2 DEEPSETS

The first architecture computes an non-weighted linear average of intermediate representations. This strategy is also known as *DeepSets* (DS) Zaheer et al. (2017). We visualize the strategy in Figure 1a.

Formally, a set representation is computed according to $\varphi_{set} = \sum_{i=1}^{n} \varphi_i$. The newly gained representation is used to predict the set utility. Again, we experiment with 2 different versions, DS-s and DS-l which compute $v = W_{|\varphi|,1} \cdot \varphi_{set}$ and $v = W_{8,1}^{(1)} \cdot \sigma(W_{|\varphi|,8}^{(2)} \cdot \varphi_{set})$, respectively.

### 3.1.3 RECURRENT NEURAL NETWORKS

The third reference architecture are recurrent neural networks, which consume object representations iteratively and generate a new representation that is supposed to represent all consumed object in each step. Contrary to DeepSets, recurrent neural networks have a more sophisticated representation aggregation function. Formally, models based on RNNs compute in each step a hidden state based on a previously computed hidden state. To increase comparability of the difference architectures,

we use the same method to compute hidden states: $h_i = tanh(W_{|\varphi| \cdot |h|, |h|}(h_{t-1} \parallel \varphi_t))$. The final hidden state is used as set representation to predict the utility of the set according to $v = W_{|h|, 1} \cdot h_n$. RNNs are visualized in Figure 1b.

## 3.2 DATASETS

To evaluate the different architectures, we conduct experiments based on visual and textual data. The datasets are described below.

### 3.2.1 MNIST REDUNDANCY AND MNIST SYNERGY

**Task Description.** The first datasets that we use are based on the MNIST dataset LeCun et al. (1998). Similar to Zaheer et al. (2017); Ilse et al. (2018), we use the images in the MNIST dataset to generate sets of number images. We assign each set a utility according to different aggregation functions. The aggregation functions are unknown to the machine learning models, whose task it is to learn to estimate the correct utility from training data.

**Set Utility.** We generate two datasets based on MNIST. The first dataset, MNIST-R, models redundancy effects. In MNIST-R, the set utility equals to the sum of all uniquely appearing digits. Hence, each element utility is only considered once and each further appearance of an already counted element utility is ignored. For example, the set $\{o_1, o_2, o_3\}$ with element utilities $o_1 = 3, o_2 = 5, o_3 = 3$ has a utility of 8 in this dataset. The second dataset, MNIST-S, models synergy effects. In MNIST-S, the utility of the set equals to the sum of all elements that appear at least twice. Elements that appear only once are ignored. The previously discussed example set $\{o_1, o_2, o_3\}$ has a utility of 3 in MNIST-S.

**Image Representation.** To represent the images, we implement and train a simple neural network model according to a publicly available digit classification model.[1] The model compresses the 784-dimensional input images in several layers to 128-dimensional and 64-dimensional representations before a softmax is applied to a 10-dimensional layer. We feed MNIST images into the trained model and extract the activations of the 64-dimensional layer, which yields 64-dimensional image representations.

### 3.2.2 SUMMARY UTILITY PREDICTION

**Task Description.** The second task used in the experiments is estimating summary utilities in automatic summarization Mani (2001); Nenkova & McKeown (2011). In automatic summarization, a set of source documents is given and the task is to generate a summary which contains the most important information from the source documents. The more important information the automatically generated summary contains, the higher the utility of the summary. Automatic summarization is often modeled as sentence selection problem in which the order of the selected sentence is not irrelevant for the evaluation. In this case, the summary is simply a set of sentences, which makes the task appropriate for the scope of this paper. Non-additive aggregation effects occur in automatic summarization naturally. Repeating the same information multiple times, for example, does not increase the information content of the summary and, hence, does not increase the summary quality. Similarly to the previous dataset, we make use of the well-known TAC2009 dataset Dang & Owczarzak (2009) for automatic summarization. The TAC2009 contains several topics, each of which contains multiple source documents. The dataset contains automatically generated summaries from over 55 systems.

**Set Utility.** Each automatically generated summary in TAC2009 is annotated with so called Pyramid SCUs (Summary Content Units) Nenkova et al. (2007). Each Pyramid SCU refers to one information nugget that is contained in the text. The SCUs have been weighted with the Pyramid method. More important SCUs have a higher weight than less important SCUs. The weight of each SCU is an integer between 1 and 4. The utility of a summary is calculated by adding the weights of all contained SCUs. However, including a SCU multiple times in a summary does not improve

---

[1]https://github.com/amitrajitbose/handwritten-digit-recognition

Table 1: Experiment results for visual and textual data

| Model | MNIST-R | | MNIST-S | | Sum-64 | | Sum-768 | |
|---|---|---|---|---|---|---|---|---|
| | MAE | MSE | MAE | MSE | MAE | MSE | MAE | MSE |
| SB-s | 5.19 | 40.60 | 7.60 | 88.71 | 1.78 | 5.03 | 0.65 | 1.02 |
| SB-l | 5.07 | 39.50 | 6.99 | 75.08 | **0.75** | 1.23 | - | - |
| DS-s | 5.20 | 40.73 | 7.62 | 89.26 | 1.78 | 5.04 | 0.49 | 0.55 |
| DS-l | 4.19 | 25.77 | 7.69 | 88.72 | 1.53 | 4.24 | - | - |
| RNN | 3.88 | 22.09 | 7.36 | 82.50 | 0.88 | 1.42 | 0.18 | 0.34 |
| RCN | 3.43 | 17.91 | 7.00 | 75.01 | 0.91 | 1.49 | **0.12** | **0.16** |
| DCR | **3.23** | **15.57** | **6.91** | **74.60** | 0.90 | **1.02** | 0.55 | 1.00 |

the utility of the summary, because repeating an SCU means that the same information is contained multiple times in the summary, which is not desired. Synergy effects are not modeled by SCUs.

**Sentence Representation.** For this task, we need a feature representation for sentences. We use two different representations in the experiments. The first representation are derived from the BERT language model Devlin et al. (2019). We use the publicly available Sentence Transformers library[2] to convert the original sentences into 768-dimensional sentence embeddings. Since the representation is high-dimensional, we also train a simple auto-encoder to generate a smaller sentence embeddings of size 64.

### 3.3 IMPLEMENTATION AND TRAINING DETAILS

We implemented all reference and newly proposed architectures in PyTorch 1.2. The source code and the used datasets will be publicly available and can be used to reproduce our experiments. All experiments have been performed on NVIDIA Tesla K80 GPUs. All datasets contain 10,000 sets. We use random splits of size 8,000, 1,000, and 1,000 for training, validation, and testing, respectively. Experiment-specific details can be found in the corresponding experiment description below. For each experiment, we limit the maximum number of epochs to 1,000 and perform early stopping based on the validation set with a patience of 50 epochs. For all experiments, we use the Adam optimizer and start with a learning rate of 0.001. We use a batch size of 100. We use the mean squared error (MSE) as well as the mean absolute error (MAE) as loss functions and evaluation metrics. Initial hidden states are initialized with zero-vectors. We use default initialization for all model weights.

## 4 EXPERIMENTS

In this section, we describe experiment-specific details, report the outcomes of the experiments, and discuss the results.

### 4.1 EXPERIMENTS WITH VISUAL AND TEXTUAL DATA

**Description.** In the first experiment, we evaluate the reference architectures and the newly proposed architectures on both visual and textual data to figure out how well the resulting models can deal with redundancy and synergy effects in both modalities. Based on the MNIST data, we generate sets with a length of 10 images and compute the set utility according to the previously described redundancy and synergy functions. For summarization, we create sets of size 4 because the average length of the summaries contained in the TAC2009 is very close to 4. Hence, set sizes of 4 are reasonable and realistic. We perform experiments with the compressed 64-dimensional and the original 768-dimensional BERT embeddings.

**Results.** To improve the quality of the experiment, we performed each experiment 5 times with different random seeds. Table 1 contains the mean of the 5 runs. We optimize for mean average

---

[2]https://github.com/UKPLab/sentence-transformers

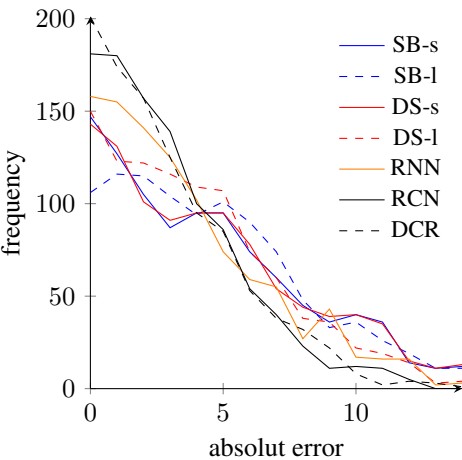
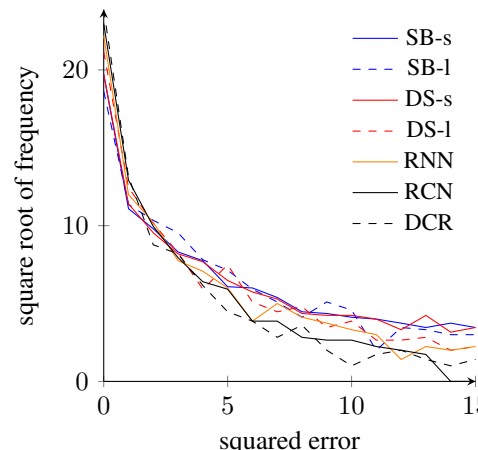

Figure 2: Distribution of absolute errors for MNIST-R experiments.

Figure 3: Distribution of squared errors for MNIST-R experiments.

error and mean squared error as described before and report results for both experiments. Overall, the newly proposed Deep Choquet Regression (DCR) architecture performs best. It achieves the lowest error in 5 out of 8 experiments. The only weakness of the architecture occurs in the summarization experiment with 768-dimensional BERT embeddings. The Recurrent Choquet Network (RCN) achieves best results in the last experiments and is placed second in all MNIST experiments. In Sum-64, it is close to the best results. With some distance, the Recurrent Neural Network (RNN) performs mediocre in the MNIST experiments. In summarization, it works better. This result might stem from the smaller set size in the summarization experiments compared to the MNIST experiments (set size of 4 vs. 10). The DeepSets (DS) and the Sum Baseline (SB) perform poorly in MNIST. SB achieves good results in Sum-64. This might be explained by the smaller impact of the redundancy effects compared to the MNIST data. In general, optimizing according to MAE and MSE correlates well. The best architecture according to MAE and MSE only differs in Sum-64. In general, we conclude that recurrent models perform better than the non-recurrent models although the input data is permutation invariant. For SB-l and DS-l, no experiments have been conducted with the large 768-dimensional BERT embeddings.

**Error Distribution.** In addition to the averages reported in Table 1, we plot the distribution of the errors in Figure 2 and Figure 3 for the MNIST redundancy experiment for the absolute error and the mean squared error, respectively. For both mean average errors and mean squared errors, it can be seen that the new models (black) make much more small mistakes and fewer large mistakes compared to the reference models (orange/blue/red). The mean squared error curves are much steeper, which results from the fact that larger errors are penalized much more than small errors.

## 4.2 VARYING SET LENGTH

**Description.** In the second experiment, we are interested in how well the architectures can deal with varying set sizes. To shed light on this question, we generate sets based on the MNIST data with redundancy and synergy aggregation functions. In contrast to the previous MNIST experiments, which have a fixed set length of 10, we randomly generate sets with lengths of 6, 8, 10, 12, and 14 MNIST images. The results of the experiments can be found in Table 2. This time, we only averaged 2 runs for each experiment due to computational limitations.

**Results.** In general, the errors tend to increase, which is reasonable due to the more difficult setup. However, the increase is rather moderate. Similar to the previous experiment, DCR and RCN perform best. This time, no clear winner can be distinguished between DCR and RCN. RNN is again placed third and the other architectures perform poorly.

Table 2: Results for varying set lengths

| Model | MNIST-R | | MNIST-S | |
|---|---|---|---|---|
| | MAE | MSE | MAE | MSE |
| SB-s | 4.93 | 38.56 | 7.77 | 91.13 |
| SB-l | 4.87 | 37.37 | 7.12 | 74.70 |
| DS-s | 4.93 | 38.08 | 7.76 | 91.07 |
| DS-l | 4.06 | 24.00 | 7.83 | 86.73 |
| RNN | 3.85 | 21.35 | 7.46 | 80.90 |
| RCN | **3.24** | **17.34** | 7.16 | 76.87 |
| DCR | 3.68 | 20.20 | **7.02** | **76.28** |

Table 3: Extrapolation results

| Model | MNIST-R | | MNIST-S | |
|---|---|---|---|---|
| | MAE | MSE | MAE | MSE |
| SB-s | 4.92 | 57.17 | 25.57 | 1292.23 |
| SB-l | 7.25 | 80.18 | 32.95 | 1350.70 |
| DS-s | 6.55 | 66.38 | **16.22** | **406.44** |
| DS-l | 6.58 | 53.21 | 37.51 | 1702.10 |
| RNN | 9.37 | 128.19 | 35.84 | 1687.71 |
| RCN | **4.23** | **29.75** | 21.74 | 866.62 |
| DCR | 4.25 | 37.79 | 16.69 | 450.81 |

## 4.3 Extrapolation to longer/shorter sequences

**Description.** In the third experiment, we are interested in the extrapolation abilities of the different architectures, i.e. how well they are able to extrapolate beyond the observed training instances. This provides a good insight in whether the models were able to identify and learn the underlying nature of the problem or if they fit to tightly to the observed distribution. To this end, we train the models on sets with length 10. Furthermore, the validation set, according to which the best model is selected, also contains only sets with length 10. The test set, however, only contains sets of size 5 and 20. Hence, the model has to extrapolate from set sizes of 10 to set sizes of 5 and 20.

**Results.** The results can be found in Table 3. Again, we report the average of 2 runs for each experiment. For MNIST-R, the RCN architecture performs best and DCR is second. The RNN architecture performs worst in the extrapolation experiment in MNIST-R. In MNIST-S, DS-s is best. This can be explained by the nature of the synergy effect. The larger the set, the more likely it becomes that images representing the same numbers appear multiple times in the set. Hence, it is not so important anymore to count the appearing digits and simply identifying the appearance of a digit can lead to reasonable results. The same effect can be observed when DS-s and RNN are compared. In prior experiments, RNN performed much better than DS-s. In this experiment, however, DS-s performs better. DCR is close to DS-s and performs second best while RCN placed third. The other architectures, namely SB-s, SB-l, and DS-l perform poorly on MNIST-S.

## 5 Related Work

**Set Representation Learning.** Most relevant to our work are works that learn representations for set regression problems. We have already discussed recurrent neural networks and DeepSets Zaheer et al. (2017). Zhang et al. (2019) learn how to order the elements of a set to feed it into an LSTM such that the resulting representation can be used to predict the desired output. Hence, it is a smart representation aggregation function. In this work, we focus on simple RNN-like hidden state aggregations in the RNN, DCN, and DCR architectures because we want to focus on whether modeling intermediate utilities can be beneficial. We leave learning of better hidden states for future work. Improving the learning hidden stated and maybe also reorder objects Zhang et al. (2019) could be interesting for future work and might improve the performance of the newly proposed DCN and DCR architectures further.

**Multiple Instance Learning.** The problem considered in this work is related to multiple instance learning Dietterich et al. (1997); Maron & Lozano-Perez (1998), and in particular multiple instance regression Ray & Page (2001). In multiple instance learning, only labels for sets of instances are observed. Usually, the problem is framed as binary classification problem in which all instances have binary labels and the set is labeled positive if at least one instance is positive. This is known as the standard multiple instance learning assumption Carbonneau et al. (2018). Similarly, Ray & Page (2001) assume that sets contain *primary* instances, which are solely responsible for the set utility. In both settings, it is sufficient to learn labels/values of the underlying instances. Furthermore, our main contributions are two new neural architectures, i.e. a contribution to representation learning, which is not considered by most prior works. Only very recently, deep learning has been applied to multiple instance classification under the standard assumption Ilse et al. (2018). In this work,

however, we focus consider problems in which a more difficult non-additive aggregation function has to be learned. Hence, learning instance utilities is not sufficient and the problem at hand is substantially different to multiple instance learning.

**Group Rating Problems.** Group rating problems relax the standard multiple instance learning assumption and do not assume the presence of a primary instance. Instead, all individuals in a group contribute to its utility. However, as already mentioned previously, most prior works in group rating assume that the set utility can be modeled as sum of the individual utilities. Most notably are the Bradely-Terry model Bradley & Terry (1952) and TrueSkill Herbrich et al. (2006). Recently, Li et al. (2017) proposed a model which learns parameters of different parametrized aggregation functions. However, their model is provided with supervision for element and group ratings. Compared to our work, we do not provide the model element scores. Furthermore, we do not provide group features. In our setting, groups are just sets of elements without any additional features.

**Sequence-to-Sequence Learning** The proposed architectures look similar to sequence-to-sequence problems such as architectures for machine translation Neubig (2017). There are, however, notable differences. First and most importantly, the predicted values in the proposed architectures are latent intermediate utilities, which are not part of the output of the prediction. Instead, the output is an aggregation of the latent utilities. In sequence-to-sequence problems, the outputs of each time step such as individual words are visible in the output such as a translated sentence. Furthermore, like in multiple instance learning, no feedback is available for intermediate predictions. Furthermore, our models predict intermediate utilities, which are numbers. Sequence-to-sequence models usually predict tokens.

**Choquet Integral for Machine Learning** The Choquet integral Choquet (1954); Grabisch et al. (2009) has been used for non-neural works before. Beliakov (2008) fit values of the discrete Choquet integral with linear programming techniques. Tehrani et al. (2012a) use the Choquet integral to model monotone nonlinear aggregations for binary classification. Tehrani et al. (2012b) use the Choquet in a pairwise preference learning scenario. To the best of our knowledge, this is the first work which combines the Choquet integral with deep learning.

## 6 CONCLUSIONS & FUTURE WORK

We found that standard methods for set regression such as architectures that belong to the DeepSets family and recurrent neural networks do not explicitly make use of the fact that meaningful intermediate utilities can be predicted in many problems. Instead, they aim at learning a representation for sets such that the set utility can be predicted solely on the learned representation. We proposed two new architectures, which have been inspired by the non-additive Choquet integral. The resulting models learn to predict and aggregate intermediate utilities. Non-additivity is essential to deal with redundancy and synergy effects.

We validated the superiority of neural non-additive utility aggregation on computer vision and summarization datasets. Besides the superior performance, the new models also enjoys a better interpretability than prior models because each prediction step is meaningful and can be inspected by laypeople that do not have any machine learning knowledge. Misconceptions of the model can be identified easily.

We focused in this paper on comparing neural non-linear utility aggregation with standard representation learning approaches for set regression problems and did not focus on achieving very high overall results. Multiple improvements of the proposed architectures can be investigated in future work. For example, more sophisticated representation aggregation functions such as LSTM and GRU cells can be used, attention mechanisms can be used to learn better hidden state representations, and different object utility functions and gate functions can be investigated. Furthermore, the proposed architectures are not limited to set regression problems, but can also be applied to sequence regression problems.

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
