# OpenReview forum: "Neural Non-additive Utility Aggregation"
_ICLR.cc/2020/Conference — Reject_

### Official Review · AnonReviewer3 · 2019-10-23
**Official Blind Review #3**

**Rating:** 3

**Review:**

This paper studies non-additive utility aggregation for sets. The problem is very interesting. Choquet Integral is used to deal with set input. The authors propose two architectures. The two architectures, though not novel enough, are towards representing “non-additive utility”.
However, the experimental comparison is not fair, the description of the model (e.g. how Choquet is integrated into the model and help to learn “intermediate meaningful results”) is not clear, some claims are not true.

First, the authors claim that they are the first to combine Choquet integral with deep learning. However, there are a few, though not many, works in the literature trying to combine Choquet integral with deep learning. For example, “Fuzzy Choquet Integration of Deep Convolutional Neural Networks for Remote Sensing” by Derek T. Anderson et al.

Second, the authors claim they are using/motivated by Choquet integral, but do not have any (appendix) sections to explain how this mathematical tool is really integrated into their models. How do you guarantee that the representation learned by the neural network still obeys the property of Choquet integral? What is your loss or your algorithm? These need to be further clarified.

Third, the comparison to baseline and “DeepSet” is not fair. According to the illustration, it seems that you first obtain “features/representations”. Then the representations are fed to the four architectures you listed in figure one. RNN-based approaches are with better “complexity” comparing to your sum baseline and “Deepset” approach. So, I have some doubts about the experimental results.


**Experience Assessment:**

I have read many papers in this area.

**Review Assessment: Checking Correctness Of Derivations And Theory:**

N/A

**Review Assessment: Checking Correctness Of Experiments:**

I assessed the sensibility of the experiments.

**Review Assessment: Thoroughness In Paper Reading:**

I read the paper at least twice and used my best judgement in assessing the paper.

---

> ### Author Response · Authors · 2019-11-13
> **Reply to Official Blind Review #3**
>
> Thanks a lot for your review and for pointing us to the reference, we will add and discuss this work in our paper. The referenced paper mimics the Choquet integral to fuse different neural networks such as CaffeNet, GoogLeNet, and ResNet50 that have been pre-trained for classification problems and can be viewed as ensemble method for multiple noisy classifiers. Contrary, we are interested in regression problems that have inherent non-additive effect such as automatic summarization. Furthermore, the referenced paper is much closer to the Choquet as we intent to be. As we describe in the paper, the proposed architectures are only inspired by the Choquet integral. This idea can be found in both of our architectures. In Figure 1c, u_i and in Figure 1d g_i * u_i model these meaningful intermediate values. We do not claim that we obtain any theoretical guarantees or properties of the Choquet integral.
>
> "How do you guarantee that the representation learned by the neural network still obeys the property of Choquet integral?"
> As described above, the proposed approaches are inspired by the way Choquet integrals handle non-additive utility aggregations. We do not claim that we obtain any theoretical guarantees or properties of the Choquet integral. Furthermore, the main idea of this work is to not learn a representation. Instead, we propose to predict many meaningful intermediate values that can simply be summed to obtain a set utility.
>
> "What is your loss or your algorithm?"
> We describe in Section 3.3 that we use mean squared error (MSE) and mean absolute error (MAE) in our experiments. We use MSE because it is usually used in regression problems. We were also interested in the mean absolute error because minimizing this loss might be more appropriate in a task such as automatic summarization, in which we don't want to punish a model strong if it makes a few severe mistakes compared to making many small mistakes. We also describe in Section 3.3. that we use Adam as optimizer.
>
> "According to the illustration, it seems that you first obtain “features/representations”. Then the representations are fed to the four architectures you listed in figure one."
> This is correct.
>
> "RNN-based approaches are with better “complexity” comparing to your sum baseline and “Deepset” approach."
> We also compare against an RNN-based approach (abbreviated with "RNN" in the paper). The RCN approach is the smallest modification one can make to implement our idea into a standard RNN. Hence, we think that the comparison is fair and meaningful. Furthermore, we demonstrate in the extrapolation experiments that standard RNNs tend to overfit. The simple sum baselines and deepsets perform better in this experiments. Hence, a "better" complexity turns out to be prone to overfitting, which shows that larger models are not necessarily better.

---

### Official Review · AnonReviewer2 · 2019-10-25
**Official Blind Review #2**

**Rating:** 1

**Review:**

This paper proposes two new architectures for processing set-structured data: An RNN with an accumulator on its output, and an RNN with gating followed by an accumulator on its output. While sensible, this seems to me to be too minor a contribution to stand alone as a paper.

Additionally, I believe the experimental tasks are new, and as a result all implementations of competing techniques are by the paper authors. This makes it difficult to have confidence in the higher reported performance of the proposed techniques.

**Experience Assessment:**

I have read many papers in this area.

**Review Assessment: Checking Correctness Of Derivations And Theory:**

N/A

**Review Assessment: Checking Correctness Of Experiments:**

I assessed the sensibility of the experiments.

**Review Assessment: Thoroughness In Paper Reading:**

I made a quick assessment of this paper.

---

> ### Author Response · Authors · 2019-11-13
> **Reply to Official Blind Review #2**
>
> Thanks a lot for your review. We address your remarks below:
>
> "RNN with an accumulator / too minor a contribution "
> We want to emphasis that the accumulator implemented in the newly proposed architectures has an inherently different nature than accumulators used so far: While accumulators such as LSTM cells accumulate knowledge about the state of a sequence, our architectures produce meaningful intermediate results, which can simply summed up to estimate the final set utility. Producing such intermediate results, which model the nature of the problem much better than previous approaches, is the key idea presented in this paper and a major benefit of the proposed architectures. This also follows the idea of the Choquet integral.
>
> "Additionally, I believe the experimental tasks are new, and as a result all implementations of competing techniques are by the paper authors. This makes it difficult to have confidence in the higher reported performance of the proposed techniques."
> To improve reproducibility, we published the data and the code. We implemented in our work the most basic version of our idea as well as the most basic version of each reference model. Hence, the code of the implemented architectures only consists of few lines and can be checked easily. We think that it is not fair to simply mistrust our results since we made our work fully transparent.

---

> > ### Comment · AnonReviewer2 · 2019-11-15
> > **baselines**
> >
> > Standard practice in machine learning is to compare against existing baselines when possible. Even with the best intentions, it is rare for authors to spend nearly as much effort carefully tuning competing methods as they do their own approach, which is why it's important to compare against previously reported performances for competing approaches.
> >
> > For instance, your work compares itself extensively against DeepSets, and the performance of DeepSets is reported on several tasks in the DeepSets paper. It might make sense to apply your method to those tasks.

---

> > > ### Author Response · Authors · 2019-11-15
> > > **Re: baselines**
> > >
> > > Dear Reviewer 2, thank you very much for your reply, we appreciate the effort.
> > >
> > > We agree that it would be great to test our work on the same datasets/tasks previous approaches such as the DeepSets approach have been tested on. Unfortunately, the problems in the DeepSets paper do not have the properties we are mainly interested in, namely problems that have redundancy and synergy effects. The reason for this situation is the generality of DeepSets: DeepSets is a general architecture that models permutation invariant sets of objects for arbitrary problems. We do not claim that our architectures are as general as DeepSets or that our architecture can compete with DeepSets on a wide range of datasets, because, as stated above, we are mainly interested in a specific type of problems that have said redundancy and/or synergy effects. Hence, using our architectures on the datasets used in the DeepSets paper is not what we want to focus on in this work.
> > >
> > > Furthermore, please note that we did not spend a lot of effort to fine-tune the results of our work. All hyper-parameters such as learning rate, optimizer, early stopping, random weight initialization, etc. are simple default choices that have not been optimized in a sophisticated way, and more importantly, the same for all models and all experiments. We report this information in Section 3.3. We will add more experimental results for different hyper-parameter settings to the appendix to show that our approaches perform consistently better than the reference systems.

---

### Official Review · AnonReviewer1 · 2019-10-30
**Official Blind Review #1**

**Rating:** 1

**Review:**

In this paper, the authors propose two RNN-based models to learn non-additive utility functions for sets of objects. The two architectures are inspired by the discrete Choquet integral. The proposed models are evaluated on visual and textual data against an MLP baseline and deep sets.

Overall, the paper is clearly written and easily understandable. However, the novelty of the paper is limited and the empirical support of the proposed models is insufficient. The motivation of using "Choquet integral" seems obscure to me. The author might want to provide a short introduction to Choquet integral and elaborate on the connection with the proposed models. The proposed models seem very basic and do not have much novelty. The tasks in the experimental study seem overly simple. The authors might want to consider other tasks, for example, Point Cloud Classification in [1].

Questions:
* For RCN and DCR, how to decide the ordering of phi_i, given that they are the objects of an unordered set?
* It would be helpful it the authors can also provide the number of parameters of the baseline models in Tables 1, 2, and 3.
* To model the interaction among objects in a set, GNN might be a better choice than RNN.

[1] Zaheer, Manzil, et al. "Deep sets." Advances in neural information processing systems. 2017.

**Experience Assessment:**

I have read many papers in this area.

**Review Assessment: Checking Correctness Of Derivations And Theory:**

N/A

**Review Assessment: Checking Correctness Of Experiments:**

I assessed the sensibility of the experiments.

**Review Assessment: Thoroughness In Paper Reading:**

I made a quick assessment of this paper.

---

> ### Author Response · Authors · 2019-11-13
> **Reply to Official Blind Review #1**
>
> We are happy that you found the paper clearly written and easily understandable. We would like to address your remarks below:
>
> "the authors propose two RNN-based models"
> While the proposed architectures have a recurrent structure, they are fundamentally different to RNNs that used to day. RNNs such as basic RNNs, LSTMs, and GRUs learn one representation of the input on which the final prediction (in our case: the set utility) is based on. No meaningful intermediate results are generated. Contrary, the proposed architectures produce meaningful intermediate results in every step and model the task at hand much better. We illustrate this fundamental difference in Figure 1. Our experiments validate that the proposed architectures perform better than standard RNNs.
>
> "The proposed models seem very basic [...]"
> We focus in this paper on the most basic versions of our idea to communicate our idea as clear as possible. We describe that multiple extensions of the idea can be investigated in future work. However, all further extensions such as more complex memory cells, which can enhance the proposed architectures, would distract from our core idea: to generate and aggregate meaningful intermediate results for set utility estimation in a non-additive way. Hence, we think the simplicity of the presented architectures is a desirable property in this work.
>
> "The proposed models [...] do not have much novelty."
> We disagree with this statement. The idea implemented in the proposed architectures has never been proposed before. We discuss [1] in our work, which is the work that is most similar to ours. Even though [1] is the most similar work, it is substantially different from ours since [1] does not use neural networks but only shallow learning. Our paper is the first that demonstrates that the idea in [1] can successfully be used with deep learning. Furthermore, the potential impact of our work is large. The problem setting we address in this paper appears in many situations. For example, a recent work [2] in automatic summarization uses the prior strategy, which we use as reference in our work. This work can potentially be improved by using non-additive utility aggregation as proposed by our work.
>
> "The tasks in the experimental study seem overly simple."
> The purpose of the experiments is to demonstrate the differences between the proposed approaches based on non-additive utility aggregation and prior ideas such as deep sets and RNNs. We show that our approaches perform substantially better than the reference approaches in computer vision and a natural language processing problems. The computer vision problem has also been used in well-known recent works [3,4]. Estimating redundancy and synergy effects of multiple sentences for automatic summarization is a well-known, hard, and unsolved problem. Hence, we think the problems used in our experiments are not overly simple but actually very hard.
>
> "The authors might want to consider other tasks, for example, Point Cloud Classification in [1]."
> Point Cloud Classification is a classification problem. However, we focus on regression problems in our work. More importantly, Point Cloud Classification is not a problem in which synergy or redundancy effects are important. Dealing with these effects is the main motivation of our work. Hence, we think that performing experiments on Point Cloud Classification is not appropriate for this work.
>
> "For RCN and DCR, how to decide the ordering of phi_i, given that they are the objects of an unordered set?"
> Training, validation, and testing examples are generated randomly and do not have a specific order. We feed all input elements, i.e. all phi_is, in the order in which they have been generated during the randomized data generation process. Hence, we use for all models the very same order. No model includes a re-ordering step. We will clarify this in the paper.
>
> "It would be helpful it the authors can also provide the number of parameters of the baseline models in Tables 1, 2, and 3."
> Thank you for the recommendation, we will add the number of model parameters to the paper. The complexity of the models can already be inspected in the published code.
>
> [1] Tehrani, Ali Fallah, et al. "Learning monotone nonlinear models using the Choquet integral" Machine Learning 89.1-2 (2012): 183-211.
> [2] Zhou, Qingyu, et al. "Neural Document Summarization by Jointly Learning to Score and Select Sentences." Proceedings of the 56th Annual Meeting of the Association for Computational Linguistics (Volume 1: Long Papers). 2018.
> [3] Zaheer, Manzil, et al. "Deep sets." Advances in neural information processing systems. 2017.
> [4] Ilse, Maximilian, Jakub Tomczak, and Max Welling. "Attention-based Deep Multiple Instance Learning." International Conference on Machine Learning. 2018.

---

### Decision · Program_Chairs · 2019-12-19

**Decision:**

Reject

**Comment:**

This paper presents two new architectures that model latent intermediate utilities and use non-additive utility aggregation to estimate the set utility based on the computed latent utilities. These two extensions are easy to understand and seem like a simple extension to the existing RNN model architectures, so that they can be implemented easily. However, the connection to Choquet integral is not clear and no theory has been provided to make that connection. Hence, it is hard for the reader to understand why the integral is useful here.  The reviewers have also raised objection about the evaluation which does not seem to be fair to existing methods. These comments can be incorporated to make the paper more accessible and the results more appreciable.